# Genomic Profiling for Piroplasms in Feeding Ixodid Ticks in the Eastern Cape, South Africa

**DOI:** 10.3390/pathogens9121061

**Published:** 2020-12-18

**Authors:** Olusesan Adeyemi Adelabu, Benson Chuks Iweriebor, Anthony Ifeanyi Okoh, Larry Chikwelu Obi

**Affiliations:** 1SAMRC Microbial Water Quality Monitoring Centre, University of Fort Hare, Private Bag X1314, Alice 5700, South Africa; AOkoh@ufh.ac.za; 2Applied and Environmental Microbiology Research Group (AEMREG), Department of Biochemistry and Microbiology, University of Fort Hare, Private Bag X1314, Alice 5700, South Africa; 3School of Science and Technology Sefako Makgatho Health Sciences University, Ga-Rankuwa, Pretoria 0204, South Africa; benson.iweriebor@smu.ac.za (B.C.I.); Lawrence.Obi@smu.ac.za (L.C.O.)

**Keywords:** piroplasms, *Theileria*, *Babesia*, ixodid, emergence, outbreak

## Abstract

Importation of tick-infected animals and the uncontrollable migration of birds and wild animals across borders can lead to geographical expansion and redistribution of ticks and pathogen vectors, thus leading to the emergence and re-emergence of tick-borne diseases in humans and animals. Comparatively, little is known about the occurrence of piroplasms in ixodid ticks in the Eastern Cape, South Africa, thus necessitating this study, which is aimed at detecting piroplasms (*Theileria* and *Babesia*) from feeding tick samples collected from cattle, sheep, and goats in selected sites in the Eastern Cape, South Africa. A total of 1200 feeding ixodid ticks collected from farm animals at selected homesteads were first subjected to molecular identification using mitochondrial 12S ribosomal RNA (rRNA) gene by PCR and were further tested for the presence of piroplasms through amplification of the 18S rRNA gene via nested-PCR followed by sequencing of the PCR products. The results indicated that 853 (71.1%) corresponded to the genus *Rhipicephalus*, 335 (27.9%) corresponded to genus *Amblyomma*, and 12 (1%) corresponded to genus *Haemaphysalis*. *Amblyomma hebraeum* and *Rhipicephalus appendiculatus* were the most common identified ticks from this study. The 18S rRNA nested-PCR revealed that 44 (3.7%) samples were confirmed positive for *Theileria*. A homology search for the generated sequences revealed a high percentage identity of 98–98.9% similarity to *T. buffeli*, *T. orientalis*, and *T. sergenti* in the GenBank. Based on the results obtained herein, we conclude that there is a big diversity of *Theileria* species; therefore, we suggest that this research should cover more geographical areas in order to reveal the true prevalence of this pathogen in the studied area because this will be a great step in the possible prevention of an outbreak that could have devastating effects on livestock production and human health in both the studied areas and South Africa at large.

## 1. Introduction

Piroplasms infections caused by pathogenic species of *Theileiria* and *Babesia* have been described as vector-borne emerging zoonoses that induce malaria-like syndrome in susceptible humans with previous exposure to infected tick bite [1,2]. Both *Theileria* and *Babesia* are haemoprotozoan that have been described as the causative agents for piroplasmoses [3,4]. They have been known to infect various species of domesticated and wild animals, causing major economic losses on livestock production, especially on small hold farmers in endemic geographical regions [5].

Parasites are transmitted to various susceptible hosts by different genera of ixodid ticks such as *Dermacentor*, *Amblyomma*, *Haemaphysalis*, *Rhipicephalus*, *Haemaphysalis*, and *Hyalomma* [2,3] during their blood meal. *Theileria* spp., as sporozoites, swiftly invade mononuclear leukocytes and thereafter develop into macroschizonts, leading to proliferation of the host cell. The genus *Theileria* has many species of which *T.*
*parva* and *T. annulata* have been reported to be the most pathogenic species of major concern to the livestock industry [6,7]. Clinical symptoms of theileriosis include fever followed by depression, nasal discharge, watery secretion from the eyes, enlarged lymph nodes, and anaemia [8] with attendant huge economic losses in endemic areas, especially Asia and Africa [9].

On the other hand, *Babesia* species are one of the most common tick-transmitted protozoan hemoparasites worldwide and have been described as the second most frequently found parasites, after trypanosomes, in mammalian blood and in a limited number of bird species. Globally, they are seen to have great medical, veterinary, and economic impacts [10,11]. *Babesia microti*, *B. divergens*, *B. duncani*, and *B. venatorum* are *Babesia* species that infect humans globally [12].

Several studies have reported the detection of piroplasms in wildlife and livestock in South Africa [13,14,15,16], but little information is available about the frequency of ixodid tick species and the prevalence of piroplasm (*Theileria* and *Babesia*) pathogens among cattle, sheep, and goats in South Africa. This study therefore reports the occurrence and genomic profiling of piroplasms in Ixodid ticks collected from domestic animals in selected homesteads in the Amatole and O.R Tambo District Municipalities, Eastern Cape, South Africa.

## 2. Materials and Methods

### 2.1. Sample Collection

With the assistance of animal health technicians and animal farm workers, 1200 feeding ticks were collected from farm animals (cattle = 718, goats = 352, sheep = 130) into sterile 50 mL Nalgene tubes containing 70% ethanol. The samples were collected between July 2017 to April 2018 in selected homesteads in the Amathole and O.R Tambo district municipalities, in the Eastern Cape of South Africa (Figure 1). Efforts were made to adhere strictly to the University of Fort Hare Animal Ethics Committee regulations on animal handling throughout the sampling period. The collected ticks were transported to the Applied and Environmental Microbiology Research Group (AEMREG) laboratory at the Department of Biochemistry and Microbiology at the University of Fort Hare for analyses. Effort was made to ensure that collected ticks from different animals and locations were properly labelled in different tubes for easy identification. Ethical clearance for the study was obtained from the University of Fort Hare research and ethics committee (Reference number: OBI013) and permission to collect samples was sought from farmers and appropriate authorities prior to sample collection.

### 2.2. Morphological Identification

Morphological identification of ticks to species level was carried out upon arrival at the laboratory using morphologic criteria such as scutum formation, capitulum formation, and limb formation [17,18]. Upon identification, the arthropods were stored at −70 °C until processed.

### 2.3. Genomic DNA Extraction

Ticks were washed in sterile distilled water 3 to 4 times for total removal of the ethanol in which they had been collected and chopped with a sterile blade in a petri dish containing phosphate buffer saline (PBS). They were then transferred into a 2 mL centrifuge tube and vortexed. After vortexing, 20 µL of Proteinase K (PK) and 200 µL of Cell Lysis buffer (CLD) were added to the homogenized samples. The tubes were then incubated at 56 °C for 2 h and centrifuged at 15,000 rpm for 1 min, and the supernatants were then aliquoted into a sterile 2 mL centrifuge tube. Engorged ticks were processed individually while non-engorged ticks of the same species were processed by pooling, taking precautions so that ticks from the same animal were pooled together using the method as described by [19]. Following this process, DNA extraction was carried out using the commercially available kit, Promega ReliaPrep^®^ gDNA Tissue Miniprep System (Madison, WI, USA), and the manufacture’s protocol was strictly adhered to.

### 2.4. Molecular Identification of Tick Species

For the molecular identification of tick species previously identified morphologically, a fragment of 338 bp of the mitochondrial 12S ribosomal DNA (rDNA) gene was amplified using a set of oligonucleotide 85F 12S [F: 5′-TTAAGCTTTTCAGAGGAATTTGCTC-3′] and 2225 12S [R: 5′ TTTAAGCTGCACCTTGAC TTAA-3′]. Polymerase chain reaction was performed in a 25 µL reaction mixture comprised of master mix; 14 μL, forward and reverse primers; 1 μL each of 10 pmol/L, 4 μL of de-ionized water and DNA template; 5 μL. The conditions used for the amplification were as followes: initial denaturation; 94 °C for 3 min; 93 °C for 30 s; denaturation, annealing; 51 °C for 30 s, elongation at 72 °C for 60 s, and final elongation at 72 °C for 5 min.

### 2.5. PCR Amplification of Tick-Borne Protozoan Pathogens

Tick-borne protozoan pathogens were screened for in all extracted genomic DNA for the presence of both *Babesia* and *Theileria* species. Two rounds of PCR with two sets of oligonucleotides targeting 18S ribosomal DNA (rDNA) gene for both species, previously reported by [20], was adopted for this study. For the first round of PCR, forward primers (5′-GGCTCATTACAACAGTTATAG-3′) and reverse primers (5′-CCCAAAGACTTTGATTTCTCTC-3′) were used to generate 930 bp while, for the second round of PCR, forward primers (5′-CCGTGCTAATTGTAGGGCTAATAC-3′) and reverse primers (5′-GGACTACGACGGTATCTGATCG-3′) were used to generate 800 bp. Three microliters of extracted genomic DNA was added to 22 µL reaction mixture comprising 14 μL of master mix, 1 μL each of 10 pmol/L of the forward and reverse primers, 4.5 μL of RNase nuclease free water, and 1.5 µL MgCl_2_, using the previously reported protocol of [20] with modification in the annealing temperature based on the Tm of the synthesis report of the primers designed by a commercial biotechnology company. For the first round of amplification, the cycling conditions follow were: initial denaturation at 94 °C for 3 min, denaturation at 93 °C for 30 s, annealing; 58 °C for 60 s, elongation; 72 °C for 1 min (40 cycles), and final elongation; 72 °C for 7 min. The same cycling conditions were used for the second round of PCR, except for an increase in the annealing temperature to 62 °C using 3 µL of PCR product from the first round of amplification.

The amplification products were analyzed in 1.5% agarose gel electrophoresis in 0.5% TBE buffer followed by staining with ethidium bromide. The gel was visualized under UV transilluminator [21]. Bi-directional sequencing was carried out on all the positive amplicons using both forward and reversed oligonucleotides that were used for PCR on the ABI3500xl automated DNA sequencer (Applied Biosystems, Foster, CA, USA) in a commercial sequencing facility.

### 2.6. Sequence Editing and Blast Search

Nucleotide sequences for both forward and reversed strands were assembled and edited to generate consensus sequences for each positive PCR product using the Geneious software program, version 10.1.2 (Biomatters, Auckland, CA, USA) [22].

The sequence data generated after editing were subjected to the BLAST program for homology searching with other curated sequences in GenBank (http://blast.ncbi.nlm.nih.gov). The search parameters were set on highly similar sequences, and sequences with a percentage similarity above 97% were downloaded for phylogenetic analysis.

## 3. Results

A total of 1200 ticks were randomly removed from domesticated ruminants (cattle, sheep, and goats) from selected homesteads from Amathole and O.R Tambo District Municipalities. Different species of ticks belonging to three genera, *Rhipicephalus*: 853 (71.1%), *Amblyomma*: 335 (27.9%), and *Haemaphysalis*: 12 (1%), in decreasing order, were identified in the present study while, at the species level (Figure 2), *Amblyomma hebraeum* showed the highest occurrence of 335 (27.9%), followed by *Rhipicephalus appendiculatus*, 274 (22.8%); *Rhipicephalus decoloratus*, 224 (18.7%); *Rhipicephalus evertsi*, 200 (16.7 %); and *Rhipicephalus microplus*, 130 (10.8%) (Table 1).

### 3.1. Detection of Piroplasms

Of the 1200 DNA samples assessed for members of the piroplasms, only 44 (3.7%) samples were confirmed positive for *Theileria*. A homology search for the generated sequences revealed a high percentage of identity above 95% with other homologous 18S rDNA reference sequences of *Theileria* spp. in GenBank. Sequences CN1 and CN2 had 96.0% homology to *T. buffeli* gene encoding 18S ribosomal RNA, *T. sergenti*-Z15106, *T. orientalis*-AB520954, *T. annulata* MF287917, and *T. sergenti*-AF162431. Subsequently, sequences CN4, CN9,CN23, CN27, CN62, CN64,CN66, CN68, CN79, CN80, and CN174 showed a higher degree of 98%–98.9% similarity to *T. buffeli* (Z15106, AB520953, AY661513, HM538196, HQ840962); *Theileria sergenti* (GU143088, EU083803, AB668373, EU083803, HM538195); *Theileria orientalis* (LC325745, AB520954, HM538223, AB668373, MH503862); *Theileria orientalis* (AB668373, AB520954); and *Theileria annulata* (MF287934, FJ225392). Other sequences, CN3, CN14, CN18, CN22, CN24, CN30, CN31, CN66, CN69, CN74, and CN76 also demonstrated a high degree of homology between 91.0%–97.0% to various *Theileria* sp. (AP011948, HM538223, EU083803, HM538220, HQ840961, KJ806987, KY197711, XR696404, LC325744, DQ286801, AB000271, KX115426).

However, homologous sequences below 90% were not reported in this study; hence, four distinct species of *Theileria* were seen to show a high degree of similarity with the generated sequences in this study.

### 3.2. Phylogenetic Analysis of Theileria Pathogen

The evolutionary history of the generated sequences were further confirmed by subjecting the derived sequence data to phylogenetic analysis with the following reference sequences of the *Theileria* 18S rRNA gene from GenBank: HM538223-*T. sergenti* (China), Z15106-*T. buffeli* (South Africa), JQ437263-*T. buffeli* (Australia), AB520953-*T.orientalis* (Australia), MF287917-*T. annulata* (India), AF162431-*Theileria* sp. (USA), MF287934- *T. annulata* (India), JQ037779-*Theileria* cf. *buffeli MC-2012* (India), HQ840964-*T. buffeli* (Scotland), U97052-*Theileria* sp. (Japan), AP011946-*T. orientalis strain Shintoku* (Japan), KJ806987-*T. buffeli* (China), FJ225391- *T. buffeli* (Spain), JQ037785-*Theileria* cf. *buffeli MC-2012* (South Africa), JX112733-*Theileria* sp. *RMP-2013* (India), GU143087-*T. sergenti* (Taiwan), AB000272-*T. buffeli* (Thailand), HM538209-*T. buffeli* (China), AY661513-*T. buffeli* (USA), KX965722-*T. buffeli* (Korea), AB520958-*T. orientalis* (Australia), GU143087 (Taiwan), JQ437263-*T.sergenti* (Australia), DQ287959-*T. buffeli* (Spain), MH327775-*T. buffeli* (Algeria), KX965721-*T. buffeli* (Korea), AB520954-*T. orientalis* (Australia), HQ840965-*T. buffeli* (Southern China), FJ426360-*T. buffeli* (Spain), KX115427-*T. sinensis* (China), JQ037788-*Theileria* cf. *sinensis MC-2012* (South Africa), LC325745-*T. orientalis* (Spain), MF287924-*T. annulata* (India), and JN572700-*Theileria* sp. B15a (South Africa). The reference sequences were previously aligned the derived sequences using BioEdit sequence alignment editor before the phylogenetic tree was generated using ClustalW in the MEGA7 database [23].

The evolutionary history was inferred using the neighbor-joining method [24]. The bootstrap consensus tree inferred from 1000 replicates is taken to represent the evolutionary history of the taxa analyzed [25]. Branches corresponding to partitions reproduced in less than 70% bootstrap replicates are collapsed. The evolutionary distances were computed using the maximum composite likelihood method [26]. Evolutionary analyses were conducted in MEGA7 [23].

Phylogenetic analysis of generated tick sequences showed that the three genera, *Rhipicephalus*, *Amblyomma*, and *Haemaphysalis*, that were initially identified through morphologic criteria clustered with respective corresponding species of the reference sequences. Sequence T21 clustered in one clade with reference sequences KC503255-*Rh. australis*, KC503261-*Rh. microplus*, KC503259-*Rh. microplus*, and AB075954-*Haemaphysalis* sp. Additionally, sequences T29, T48, and T29 clustered closely with reference strains AF031859-*Rh. appendiculatus*, DQ801282-*Rh. appendiculatus*, KX276945-*Rh. appendiculatus*, DQ849237-*Rh. zambeziensis*, DQ849224-*Rh. zambeziensis*, MF3611814-*Rhipicephalis* sp., and MF479197-*Rh. appendiculatus*. Likewise, sequences T01, T4, T13, T20, T25, and T32 clustered together in a clade with reference sequences KF583637-*H. longicornis*, AF31853-*H. longicornis*, HQ434625-*H. longicornis*, AF150049-*A. hebraeum*, MG076932-*A. maculatum*, AY342288-*A. triste*, AY342261-*Amblyomma* sp., KT386309-*Amblyomma* sp., and AF150049-*A. hebraeum*. In addition, obtained sequence T40 revealed high closeness to reference sequences MF479198-*Rh. Evertsi*, AF150052-*Rh. Evertsi*, and MF348105-*Rh. evertsi*, while sequence T10 was found to be closely related to KY676830-*Rh. australis*, EU9217770-*Rh. microplus*, AF150045-*B. annulatus*, and AF031847-*B. microplus* (Figure 3).

The evolutionary history was inferred using the neighbor-joining method [24]. The optimal tree, with the sum of branch length = 0.96781808, is shown. The evolutionary distances were computed using the p-distance method [27] and were in the units of the number of base differences per site. Evolutionary analyses were conducted in MEGA7 [23].

Phylogenetic analysis revealed that all the *Theileria* sequences obtained in this study clustered into four clades as the majority of the generated sequences clustered unambiguously with each other. Thirteen sequences clustered in a clade with reference sequences HM538209-*T. buffeli* and MF287924-*T. annulata*, while two other sequences (CN62 and CN84) were found to cluster between reference sequences MF287924-*T. annulata* and U97052-*Theileria* sp. Interestingly, sequences CN1, CN85, and CN21 were found to cluster phylogenetically with reference strains FJ225391-*T. buffeli* and MF287934-*T. annulata*, while CN72 and CN125 clustered between LC325745-*T. orientalis* and HQ840964-*T. buffeli* (Figure 4).

### 3.3. Accession Number

The nucleotides sequences generated from this study were submitted to GenBank under the following accession numbers: MK347068–MK347111 for *Theileria*, while the representative sequences were deposited for *Amblyomma hebraeum*, *Rhipicephalus microplus*, and *Haemaphysalis longicornis* under accession number MK347205–MK347212.

## 4. Discussion

*Theileria and Babesia* are the two major genera of the Piroplasmorida group belonging to the phylum *Apicomplexa.* They are tick-borne intracellular haemoprotozoan parasites and have been described to be of global economic, veterinary, and medical significance [11,28]. Although human infection due to the theileria pathogen has not been reported, several studies have described human infections caused by *Babesia* species [29,30,31]. In this study, only 44 (4%) of the DNA samples were confirmed positive for *Theileria* pathogens, while none were positive for *Babesia*. Five distinct species of *Theileria* (*T. sergenti*, *T. buffeli*, *T. orientalis*, *T. annulata*, and *T. sinensis*) were seen to show a high degree of homology to the reference sequences obtained from GenBank.

Different literature has described the *T. sergenti*, *T. buffeli*, and *T. orientalis* groups to be benign bovine protozoan species responsible for major economic losses in the livestock industry and, owing to the high similarity in their serological and morphological cross-reactivity, they are now referred to as *T. orientalis* [32,33]. The pathogenicity and infectivity of each *Theileria* species have been seen to usually differ from one geographical region to another. For instance, *T. buffeli* has been seen to be of minor impact in cattle compared to *T. parva* [34].

Different species of *Rhipicephalus*, *Amblyomma*, and *Haemaphysalis* have been reported as being vectors of *Theileria* pathogens among domestic and wild animals [35,36]. A prevalence of *Theileria* pathogens among domestic animals has been reported from different geographical regions of the world such as France [37], Thailand [38], China [39], and Australia [40], as well as in Japan, Italy, Greece, Iran, Ethiopia, and India [41,42,43].

In Australia, it has been reported that, among the *Theileria* spp., *T. orientalis* has been a major concern for several years, especially among cattle, owing to the spread and emergence of the pathogen from wild animals; it has cost the Australian government over $19 million yearly, yet no vaccines or chemotherapeutic treatments are available [44]. Likewise, the occurrence of theileriosis caused by *T. annulata* has been described as being extensive in subtropical regions in the Northern Hemisphere, extending from northern and southern Africa, Europe via the Middle East, and Asia, and it causes an acute theiloriosis similar to tropical theileriosis by *T. parva*. European breeds of cattle are more susceptible to the infection and suffer a high degree of mortality [33].

Similarly, *T. orientalis* was reported to be the cause of an outbreak in Australia and New Zealand, with occurrences in a large number of herds and severely affected dairy farms with clinical signs such as haemolytic anaemia, fever, frequent death among the animals, increases in abortion and stillbirths, and drastic reduction in milk productions being observed in most cases [45,46,47,48,49].

Additionally, *T. orientalis* has been implicated in the outbreak of bovine theileriosis in Burundi; morphological and serological analyses were used to establish its identity, and *Amblyomma variegatum* was reported as the local tick vector [50]. This study has revealed, for the first time, the occurrence and presence of *T. orientalis* in southern Africa, as previous studies only described the presence of *T. parva* from both domestic and wild animals [13,51,52].

Through conventional PCR, a high prevalence of *T. orientalis* has been reported in cattle from different Asia-Pacific regions including Mongolia, 41.7% [38]; Sri Lanka, 53.5% [5]; Thailand, 31.5% [53]; Vietnam, 13.8% [32]; Myanmar, 36.2% [54]; and Japan, 64.8% [55], compared to the 4% prevalence rate shown in this study, which is however higher than the 0.68% recently reported in cattle from Egypt [56]. Different factors could be responsible for the variation in the prevalence of *T. orientalis* in different geographical regions, which include the existence and abundance of ticks, susceptibility of animal breeds, wildlife reservoirs, and other environmental or climatic factors.

A free grazing method was observed among the farm animals in the study areas, and this could serve as a perfect platform for the transmission of *T. orientalis* pathogens from infected tick vectors from the wild. The abundance of African buffalo, being the reservoir host of tick vectors, has been reported in southern Africa, especially the free-ranging African buffalo that usually graze on the same vegetation as farm animals, especially in the study areas [52].

Another causative agent of tropical theileriosis detected in this study was *T. annulata*, second to *T.parva* among *Theileria* spp. of high pathogenicity and major economic importance in livestock productions, usually transmitted by *Hyalomma* ticks. Tropical theileriosis caused by *T. annulata* is mostly present in southern Europe and northern Africa, spreading through the Middle East, India, and southern Russia into China [57]. High mortality and morbidity rates have been reported in both exotic and indigenous breeds, thus leading to devastating economic losses in livestock production, especially in developing countries [58,59].

The prevalence of theileriosis caused by *annulata* pathogens has previously been reported in different countries; a prevalence rate of approximately 30% has recently been reported in Portugal [60], about 76.87% has been reported in Pakistan [61], 23% in India [62], 64% in Egypt [63], 18% in China [64], and 24% in Spain [65]. There is a dearth of information on the occurrence of *T. annulata* in South Africa, except for the report of [66], who reported its occurrence in the country.

Phylogenetic analysis from this study showed that *T. annulata* clustered unambiguously with other reference strains from India, Korea, and Spain; hence, it could be hypothesized that the presence of *T. annulata* in the studied areas is as a result of migratory wild animals and birds carrying infected ticks with them. *Hyalomma* ticks have been described as the main vectors of *T. annulata* worldwide; however, species of *Haemaphysalis*, which are also endemic in the studied areas, are recently being implicated in its transmission in the Philippines [67].

The universal 18S rRNA gene is commonly reported and used for deducing the phylogeny of different eukaryotic organisms as well as for delineating the species and strains/genotypes of *Theileria* and *Babesia* species [16,66,68]. It has been used to identify various distinct genotypes of *Theileria* in various reservoir hosts worldwide, including ruminants and wild animals [53,56,69,70]. Phylogenetic analysis of the derived sequences based on the 18S rRNA gene in this study showed that all the *Theileria* sequences clustered into four clades, which suggests that the obtained sequences could be polymorphic in nature. *Theileria* sequences form this study showed a high genetic similarity to those from Spain, Korea, Thailand, and Japan, which is supported by the findings of [64].

## 5. Conclusions

This study has revealed the presence of *T. annulata* from ticks collected from domesticated animals for the first time in the studied areas. Based on the results obtained herein, we conclude that there is a big diversity of *Theileria* species, therefore we suggest that further research should be conducted that will cover more geographical areas. It would also be good to test the animals for the presence of theilerial pathogens in order to establish the true prevalence and population dynamics of ticks throughout the year and to determine the seasonality of this pathogen in the studied area. This would be a great step in the possible prevention of an outbreak that could have a devastating effect on livestock production in both the studied areas and South Africa at large.

## Figures and Tables

**Figure 1 pathogens-09-01061-f001:**
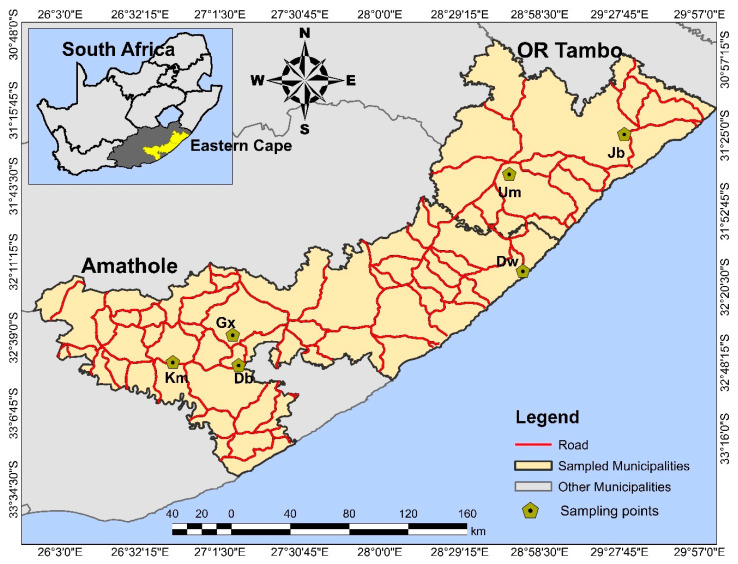
The map showing the geographical locations of the sampling sites with their coordinates; Debe (Db) = 32°52′11.852″ S, 27°1′14.171″ E; Gxulu (Gx) = 32°40′26.702″ S, 27°6′19.591″ E; KwaMemela (Km) = 32°47′38.497″ S, 26°44′10.889″ E; Dwesa (Dw) = 32°13′50.916″ S, 28°51′16.135″ E; Umtata (Um) = 31°39′26.69″ S, 28°48′0.194″ E; Jambini (Jb) = 31°23′36.856″ S, 29°29′46.921″ E. Map created using ArcMap 10.5.1.(ArcGIS, Esri, Redlands, CA, USA)

**Figure 2 pathogens-09-01061-f002:**
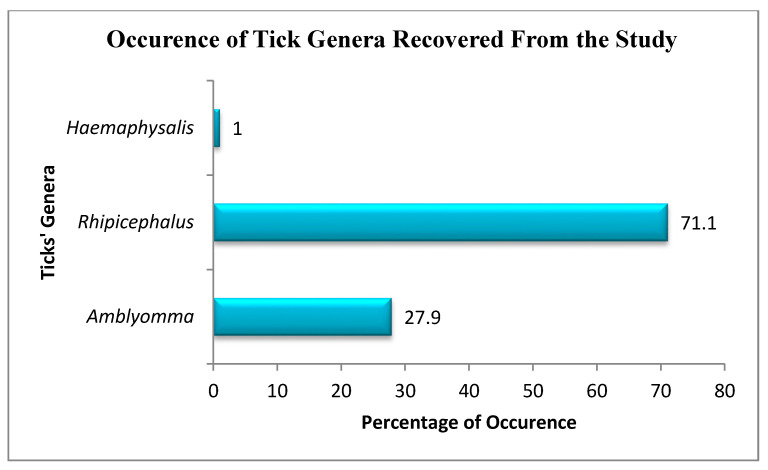
The prevalence of tick genera collected in the study. The figure shows the overall prevalence of tick genera collected at all the sampling sites.

**Figure 3 pathogens-09-01061-f003:**
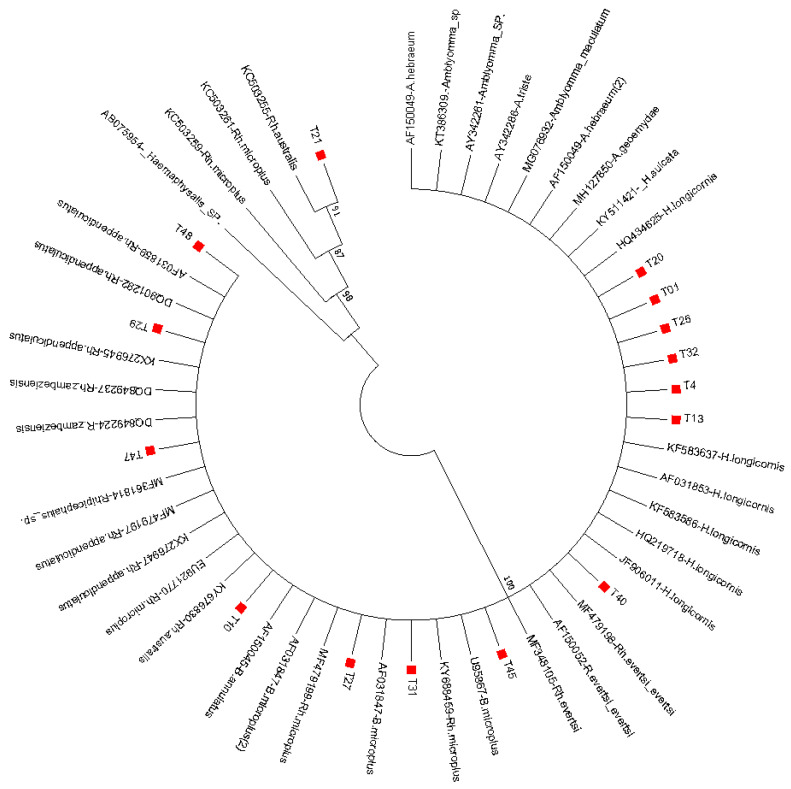
Evolutionary relationships of tick species based on nucleotide sequences of mitochondrial 12S ribosomal RNA gene.

**Figure 4 pathogens-09-01061-f004:**
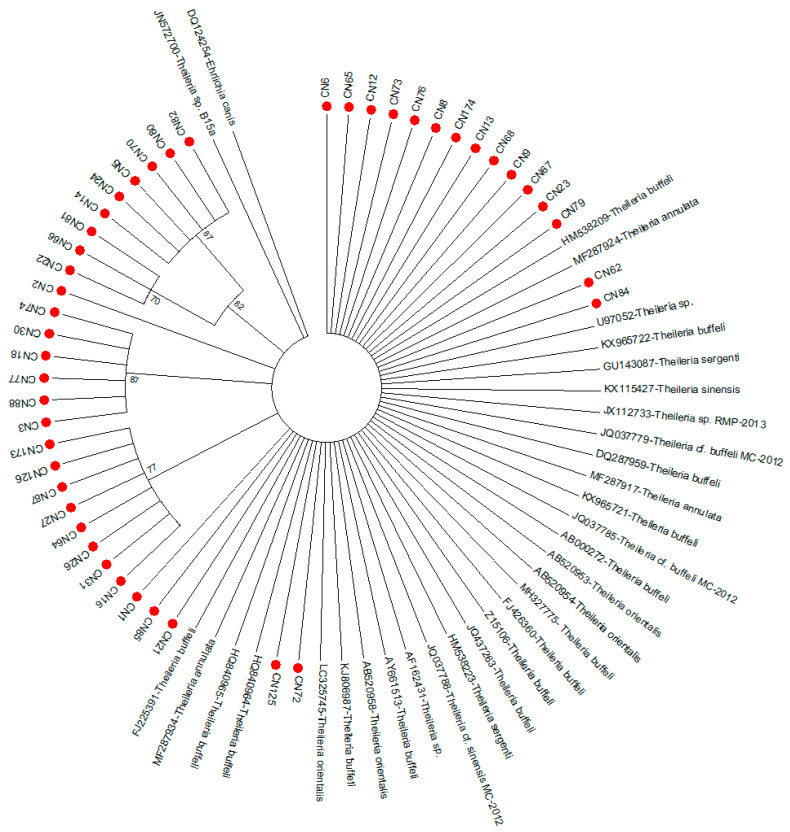
Evolutionary relationships of different *Theileria* spp. using nucleotides sequence of the 18S ribosomal RNA (rRNA) gene.

**Table 1 pathogens-09-01061-t001:** Diversity of Tick Species Collected from the Animals in the Study Areas.

Tick Species	Number of Tick Species per Animal			Total Number of Ticks (%)
	Cattle	Goat	Sheep	
*A. hebraeum*	235	80	20	335 (27.9)
*Rh. decoloratus*	129	70	25	224(18.7)
*Rh. sanguineus*	0	15	5	20 (1.7)
*Rh. eversti eversti*	140	40	20	200 (16.7)
*Rh. microplus*	70	40	20	130 (10.8)
*Rh. appendiculatus*	139	95	40	274 (22.8)
*Rh. zambeziensis*	5	0	0	5 (0.4)
*H. spinulosa*	0	12	0	12 (1.0)
Total	718	352	130	1200

*Rh.* = Rhipicephalus, *A.* = Amblyomma, *H.* = Haemaphysalis.

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
