# Peer review of "Genomic Profiling for Piroplasms in Feeding Ixodid Ticks in the Eastern Cape, South Africa"

_pathogens, 2020, doi:10.3390/pathogens9121061_

Round 1

Reviewer 1 Report

The manuscript 1027722 with the full title of " Genomic profiling for piroplasms in feeding ixodid ticks in the Eastern Cape, South Africa" has been submitted. It is an interesting study. The authors presented a study of epidemiology of piroplasms in feeding ixodid ticks. 1200 feeding ixodid tick samples were collected and detected by PCR and nested-PCR. Furthermore, the authors sequenced the PCR products and generate the phylogenetic analysis based on the sequences.

Overall, the manuscript seems to be well organized and concisely written to me. However, there are few concerns need to be addressed: 

1) Figure 1, the authors could give a big picture of the locations, such as a map of South Africa, where the samples were collected, which would give a better idea to the readers.

2) Page 2, Line 42, please modify the font of “Amblyomma”

3) Page 10, line 242, please modify the font of the sentence “Theileria and Babesia are the two major genera of Piroplasmorida group belonging to the phylum Apicomplexa.”

4) Page 10, line 251, please modify the font of “Different literatures have described”. Please also check the entire ms.

Author Response

The response to reviewer's comments has been attached.

Reviewer 2 Report

  • Add citation to this sentence “The genus Theileria has many species of which parva and T. annulate have been reported to be the most pathogenic species of major concern to livestock industry”
  • “0–41.5 °C,” is “0” body temperature correct?, pls check
  • Update the following citation “Globally, they have been described to be of great medical, veterinary economic and impacts (Schnittger et al., 2012)”, there are many more recent citation for this sentence
  • Pls add approval number for your ethical approval and clarify the approval form owners was written or oral?
  • The authors mentioned that “Engorged ticks were processed individually while non-engorged ticks of the same species were processed by pooling taking precaution so that ticks from same animal are pooled together using method as described by James et al. (2014)”, can the authors give more details about the number of farm animals (cattle, goats and sheep) from which one thousand two hundred feeding ticks were collected in the M&M section?, also please mention the exact number of each animal spp. (cattle, goats and sheep) from which the ticks was collected?
  • Pls give an acceptable explanation to the following ticks decreasing order; Rhipicephalus: 853 (71.1%), Amblyomma: 335 (27.9 %) and Haemaphysalis: 12 (1%)?
  • L244, “(Schnittger et al., 2012)”, update pls.
  • Pls check this citation “Elsify et al., 2015” is mentioned in the reference list. Pls check all the references that mentioned in the text are presented in the reference list accordingly.

Author Response

Kindly find attached the response to the reviewer's concerns.
